# Variation in the Health Status of the Mediterranean Gorgonian Forests: The Synergistic Effect of Marine Heat Waves and Fishing Activity

**DOI:** 10.3390/biology13080642

**Published:** 2024-08-21

**Authors:** Martina Canessa, Rosella Bertolotto, Federico Betti, Marzia Bo, Alessandro Dagnino, Francesco Enrichetti, Margherita Toma, Giorgio Bavestrello

**Affiliations:** 1Dipartimento di Scienze della Terra dell’Ambiente e della Vita—DISTAV, Università di Genova, Corso Europa 26, 16132 Genova, Italy; federico.betti@edu.unige.it (F.B.); marzia.bo@unige.it (M.B.); francesco.enrichetti@edu.unige.it (F.E.); margherita.toma@isprambiente.it (M.T.); giorgio.bavestrello@unige.it (G.B.); 2Agenzia Regionale per la Protezione dell’Ambiente Ligure—ARPAL, Via Bombrini 8, 16149 Genova, Italy; rosella.bertolotto@arpal.gov.it (R.B.); alessandro.dagnino@arpal.liguria.it (A.D.); 3Consorzio Nazionale Interuniversitario per le Scienze del Mare, Piazzale Flaminio 9, 00196 Roma, Italy; 4National Biodiversity Future Centre—NBFC, Piazza Marina 61, 90133 Palermo, Italy; 5Istituto Superiore per la Protezione e la Ricerca Ambientale—ISPRA, Via Vitalino Brancati 48, 00144 Roma, Italy

**Keywords:** structuring alcyonaceans, marine strategy framework directive, anthropic impact, global change, conservation, Ligurian sea

## Abstract

**Simple Summary:**

Epibiosis on damaged gorgonians, particularly *Paramuricea clavata*, is generally used as an indirect indication of stressed conditions. Studying damaged colonies over time can help us understand the temporal evolution of the phenomenon. Positive thermal anomalies recorded before 2019 were identified as the primary cause of damage to gorgonians. However, the decrease in epibiosis percentages after 2019, despite an episode in 2022, may be related to the resilience of the populations and a possible reduction in fishing activities due to the lockdown imposed during the COVID-19 pandemic. Long-term monitoring programmes are fundamental in clarifying the trajectories of changes in marine benthic communities exposed to different stressors.

**Abstract:**

Over the past thirty years, the red gorgonian *Paramuricea clavata* in the Mediterranean Sea has faced increasing threats, including heat waves and human activities such as artisanal and recreational fishing. Epibiosis on damaged gorgonian colonies is generally used as an indirect indication of stressed conditions. The density and height of *P. clavata* and the percentage of colonies affected by epibiosis and entangled in lost fishing gear were monitored to investigate the phenomenon and its trend over time in the Ligurian Sea. Analyses were based on transects collected during ROV campaigns between 2015 and 2022 at depths of 33–90 m. A strong correlation was observed between fishing efforts in the study area and the level of epibiosis. Maximal percentages of colonies affected by epibiosis and entanglement were recorded at depths of 50–70 m. Temporally, marine heat waves before 2019 were identified as the primary cause of damage to *P. clavata*. The decrease in epibiosis percentages after 2019, despite the 2022 heat wave, may be due to a quick recovery ability of the populations and a reduction in fishing activities during the COVID-19 lockdown in 2020. Long-term monitoring programmes are essential to understand the changes in marine benthic communities exposed to different stressors.

## 1. Introduction

The red gorgonian *Paramuricea clavata* (Risso, 1827) (Cnidaria: Anthozoa) is among the main habitat-forming species in the Mediterranean circalittoral zone, and its aggregations are known to heavily influence the diversity and structure of the associated assemblage [1,2,3]. *P. clavata* forests constitute noteworthy seascapes attracting SCUBA diving tourism, thus resulting in an important economic resource [4,5]. Moreover, they host significant populations of commercial fish such as the common dentex *Dentex dentex* (Linnaeus, 1758) and the dusky grouper *Epinephelus marginatus* (Lowe, 1834), typical targets of artisanal fishing [6].

*P. clavata* forests can be impacted by different kinds of pressures, mainly related to global warming and anthropogenic activities; therefore, the species is considered “Vulnerable” according to the Red List of the International Union for Conservation of Nature (IUCN), and the reduction in some Mediterranean populations has been assessed in the last three decades [7]. The first pressure is related to the recent increase in positive thermal anomalies causing cascade effects such as the development of pathogenic bacteria and blooms of filamentous algae and mucilage. In turn, these phenomena affect the gorgonians with extensive diseases that favour settling invertebrate epibionts on the damaged colonies [8,9,10,11,12,13,14,15,16]. Marine Heat Waves (MHWs) also result in Mass Mortality Events (MMEs) that have been well-described for many organisms of the coralligenous assemblage [11,17,18,19,20,21,22,23], but mainly hit the red gorgonians [24,25,26,27].

In addition to climate-related impacts, it is well known that boat anchoring and demersal recreational and artisanal fishing activities threaten *P. clavata* due to its large size, branched shape, and skeleton limited flexibility that favour the entanglement of fishing gear [20,28,29]. Bavestrello et al. [28] described the abrasive action by lost lines in contact with the gorgonian coenenchyme, producing suitable conditions for the settlement of several epibiotic organisms that ultimately cause the total or partial breaking of the colony under the current flow. In this regard, several authors employed the percentage of epibionted colonies as a biological index quantifying the fishing-induced stress of a gorgonian population [24,30,31,32,33,34,35,36,37,38]. Variable levels of epibiosis, and hence the health status of the populations, have been reported in different Mediterranean sites. A study conducted in the Medes Islands (Catalan Sea) indicated that 10–33% of the colonies settled in unprotected areas were partially colonised by epibionts, whereas only 4–10% of the populations in a site of the Marine Protected Area (MPA), where fishing is prohibited, were colonised [39]. Along the eastern Adriatic coasts, in an area characterised by low fishing pressure, the colonies with denuded or epibionted branches were less than 10% [40]. On the contrary, inside the Portofino MPA (Ligurian Sea) and in the Tavolara—Punta Coda Cavallo MPA (Tyrrhenian Sea) colonies with epibiosis exceeded 50% of the total, suggesting a high fishing impact [6,41,42].

Monitoring programs on a broad spatial and temporal scale dedicated to evaluating the health status of gorgonian forests are lacking, and the recovery ability of populations after mass mortality episodes were only followed for a few years [43,44,45,46]. The Marine Strategy Framework Directive (MSFD 2008/56/EC) represents the EU’s Integrated Maritime Policy tool to achieve the Good Environmental Status (GES) of marine waters, based on eleven qualitative descriptors. Within these, the descriptor “biodiversity” states that “*The quality and occurrence of habitats and the distribution and abundance of species are in line with prevailing physiographic, geographic and climatic conditions (MSFD, 2008/56/EC, Annex I). Ecosystem components—groups of species and habitat types, such as the coralligenous—are recognised as indicators of environmental quality, becoming priority habitats target of study and monitoring activities*”.

Due to the data obtained during the Marine Strategy project (2015–2022), the benthic communities of the Ligurian deep continental shelf and shelf break (NW Mediterranean Sea) are currently one of the best-known areas along Italian coasts [47]. During this project, 80 sites were explored (2015–2018) and then subjected to monitoring in the following years. The forests of *P. clavata* were widely distributed along the whole Ligurian arc, occurring on sloping outcroppings and coralligenous reefs. The forests dominated by this species represented over 13% of the sampling units considered in the study and were characterised by an average density of 7.4 ± 3.0 colonies m^−2^ with maximum values reaching 19 colonies m^−2^. This gorgonian also consistently occurred in other communities, such as that dominated by *Corallium rubrum* (Linnaeus, 1758) [47].

Thanks to the large ROV footage collected during the Ligurian Marine Strategy surveys, this paper aims to give a comprehensive view of the level of epibiosis affecting the *P. clavata* forests along the whole Ligurian coastline, test for putative correlation between gorgonian health status and fishing effort and other related drivers and describe a pluriannual trajectory of their health status evolution.

## 2. Materials and Methods

### 2.1. Study Area

The Ligurian coast extends for over 350 km, from La Spezia in the East to Ventimiglia in the West, in the North-western sector of the Mediterranean Sea (Figure 1). The topographic setting along the coastline is very different: in the eastern part, from La Spezia to Genoa, the continental shelf is wide and gently sloping, while in the western sector, from Genoa to Ventimiglia, it is deeper, steeper, and narrower, incised by several submarine canyons [48]. In the stretch of coast within the province of Savona, the coastline is oriented towards the SE, while in the province of Imperia, it is bent towards the S. The whole Ligurian water circulation moves westward, driven by the overall cyclonic circulation of the basin [49,50]. The coastal zones are densely populated with cities and large commercial harbours; artisanal small-scale fisheries and recreational activity are widespread, especially in the summertime [51]. In this study, the Ligurian coast was divided into three areas based on physiographical characteristics (depth of continental shelf and coast exposition) according to Canessa et al. [52]: East (A1), from Mesco Cape to Isuela Shoal (Portofino Promontory); Centre (A2), from Arenzano to Finale Ligure; and West (A3), from Albenga to Mortola Cape (Ventimiglia) (Figure 1; Table 1).

Data from the European Fleet Register (https://webgate.ec.europa.eu) were used to obtain the number of fishing harbours and boats targeting demersal fishing, focusing on each identified area. The water quality along the Ligurian coast, evaluated through the CARLIT Index, was available on the Site of Liguria Region (https://ambientepub.regione.liguria.it/SiraRsaFruizionePubb/IndicatoreRsa.aspx?page=1&Anno=2017&Codtrel=RSA&Sezione=8&Riga=6&FlagQuadroTot=0) (accessed on 26 June 2024).

### 2.2. ROV Footage Analysis

The surveys were carried out following the guidance reported on the Marine Strategy Framework Directive (MSFD) methodological sheet created in 2016 by the Ministero dell’Ambiente e della Tutela del Territorio e del Mare (MATTM) in collaboration with ISPRA and ARPA, relating to coralligenous habitats (Module 7, Rocky reefs), including all following modifications (Art. 11, D.lgs. 190/2010). From 2015 to 2022, 264 transects were performed in the 33–90 m depth range along the whole Ligurian coast and exactly repeated to check putative differences after a span of time of 2–4 years.

Each ROV transect had a length of 200 m and a width of 50 cm, covering 100 m^2^ of the seafloor, as required by the MSFD protocol for deep coralligenous monitoring; the technical specifics of ROVs, tracks, and video analysis used for transects are available in Enrichetti et al. [34].

In each transect, all of the colonies of *P. clavata* completely framed by the camera were counted, and their density was calculated, referring only to the percentage of hard bottom seafloor present in the transect, thus excluding soft bottom areas unsuitable for the settling of the gorgonian.

Sessile epibionts (sponges, anthozoans, bryozoans, annelids) and vagile fauna spotted associated with the colonies were noted. The percentage of colonies with epibiosis or entangled in fishing gear was used to estimate the health status of the populations. The height of 100 colonies randomly selected per transect was measured using the laser pointers as a reference (15 cm apart). In transects with less than 100 gorgonians, all of the colonies were measured (Table 1).

Only the transects with more than 50 colonies were selected for the formal analysis. In particular, we used 13 transects replicated after 3–4 years in A1, 19 transects replicated after 3 years in A2, and 14 transects replicated after 2–3 years in A3 (Table 1).

Considering the entire database, putative differences in the percentage of epibionted colonies out of the total measured were investigated among areas and compared with the number of fishing harbours and fishing vessels working with set gear (set longlines, set gillnets, and trammel nets). Differences across time (before/after) were studied within each area, comparing data of epibiosis and entanglement in the same transects replicated after 2–4 years.

To assess whether the putative trajectories of temporal variations were generalised at a spatial scale or linked to stochastic phenomena, we evaluated the percentage of transects showing a positive (epibiosis/entanglement increasing) or negative (epibiosis/entanglement decreasing) change in the percent of affected colonies during each considered period.

### 2.3. Statistical Analysis

Differences in *P. clavata* density, colony height, and percentage of epibiosis in the whole study area were tested using one-way ANalysis Of VAriance (ANOVA). Temporal analysis of the replicated transects was performed using PERMutational ANalysis Of VAriance (PERMANOVA) [53] (factor “area”, fixed, 3 levels, and “time”, i.e., before/after, fixed, 2 levels; Bray–Curtis similarity Index measure, permutation = 9999). All statistical analyses were performed using PRIMER-e version 7.0 with the PERMANOVA+ Add On package.

## 3. Results

### 3.1. Health Status of Paramuricea Clavata Forests in the Ligurian Sea

In total, 24834 colonies of *Paramuricea clavata* were observed, of which 6836 were affected by epibiosis (28%) and 6140 were entangled in lost fishing gear (25%) (Figure 2).

The forests appeared to be homogeneously distributed along the Ligurian arc (Figure 1). The average density (about 3.3 ± 0.5 colonies m^−2^) showed no significant differences among the three areas, as well as according to depth (Figure 3a,b; Table 2). The mean height was significantly lower in A2 (26.6 ± 1.5 cm) than in the two other areas (29.4 ± 1 cm and 31 ± 1.5 cm, respectively, in A1 and A3) (Figure 3c; Table 2). Particularly, the percentage of the epibionted colonies significantly decreased from the eastern area (A1, 36%) to the centre (A2, 30%) and the western ones (A3, 18%) (Figure 3d; Table 2).

This trend was correlated with the fishing effort estimated per each area; in fact, the number of fishing harbours was 11, 7, and 3, respectively, in A1, A2, and A3, and the fleets were represented by 121 vessels in A1, 136 in A2, and 98 in A3 (Figure 4a,b). An inverse correlation could also be observed with values of the Carlit index that showed a situation from sufficient to very good for A1 and A2, while it showed from good to very good for A3 (Figure 4c).

The bathymetric trend in the incidence of the epibiosis showed maximal values in the depth range 50–70 m (32–35% of the observed colonies), with a decrease at lower and higher depths (Figure 5a). The bathymetric distribution of the colonies entangled in lost fishing lines reflected a very similar trend (Figure 5b).

The species most frequently found as epibionts on *P. clavata* were the sponge *Pleraplysilla spinifera* (Schulze, 1879), the soft coral *Alcyonium coralloides* (Pallas, 1766), the bushy serpulids belonging to the *Filograna/Salmacina* complex, and the branched bryozoans *Adeonella calveti* Canu & Bassler, 1930, *Smittina cervicornis* (Pallas, 1766), and *Turbicellepora avicularis* (Hincks, 1860). Finally, specimens of the passive filter-feeder echinoderm *Astrospartus mediterraneus* (Risso, 1826) were commonly observed.

### 3.2. Temporal Analysis of Epibiosis and Entanglement

A comparison of the same groups of transects replicated after 2–4 years revealed a similar trend in the three considered areas.

In area A1, the transects studied in 2015 and replicated in 2019 showed a significant increase in the percentage of epibionted colonies, shifting from 14% to 61%. A similar trend was observed in the transects made in 2016 and replicated in 2019, with values increasing from 13.8% to 35.9%. In the group of transects studied during 2019 and repeated in 2022, the incidence of epibiosis was not significantly different (Figure 6a; Table 3). These variations were homogeneously recorded among the considered transects, confirming the progressive shift. In fact, in the periods 2015–2019 and 2016–2019, almost all of the studied transects showed positive variations (increase) in the epibiosis. Conversely, in the period 2019–2022, 80% of the transects showed negative percentage variations (decrease) (Figure 6a inset). The percentage of entangled colonies significantly surged from 2.5% to 38.9% in the period 2015–2019, while in both the following periods (2016–2019; 2019–2022), no significant differences were assessed (Figure 6b; Table 3). The trend in the percentage of transects with positive/negative variations showed 100% positive variations in the first period, while in the following periods, the percentage of negative variations progressively grew (Figure 6b inset).

A similar situation was observed for area A2. A comparison of the transects carried out in the periods 2015–2018 and 2016–2020 showed a progressive steady rise in the percentage of colonies affected by epibiosis, from 8.7% to 47% and from 4.6% to 43.5%, respectively. In the period 2018–2022, no significant differences were detected (Figure 7a; Table 3). In the first two periods, the totality of studied transects showed soaring values, while in the third period, this value decreased to 50% (Figure 7a inset).

Regarding the entanglement percentages, a strong increase was observed in the first period, followed by a period without significant variations, and, finally, a third period was characterised by a significant decrease (Figure 7b; Table 3). An analysis of the homogeneity of the variations across time showed a regular trend from 100% of transects with positive variations in the period 2015–2018 to 100% of transects characterised by a negative variation in the period 2019–2022 (Figure 7 inset).

Finally, regarding area A3, a diachronic analysis highlighted a significant soaring of the epibionted colonies in the first period, followed by two periods without significant variations (Figure 8a, Table 3). The corresponding trend in the percentage of transects with positive variation progressively decreased from 100% in the first period to 33% in the third one (Figure 8a inset). Finally, the trend in the entanglement showed a significant increase in the first period followed by a significative decrease in the other two periods (Figure 8b; Table 3), also confirmed by a specular trend in the percentage of transects with positive/negative variations (Figure 8b inset).

## 4. Discussion

The diseases affecting the gorgonian populations in the Mediterranean Sea are generally attributed to the acute stress produced by positive thermal anomalies that have progressively increased in frequency and intensity in recent decades [22,46,54]. During the period of our observations, MHW events were recorded in 2015, 2016, 2018, and 2022 [22,55].

Our study revealed that the epibiosis affecting *Paramuricea clavata* in three areas along the Ligurian coast had an overlapping temporal trajectory that can be generalised at the regional scale despite the geomorphological differences in the littoral portions and the presence of local anthropic pressures such as large commercial harbours.

The percentage of affected colonies strongly increased from 2015 until 2019, while in the following period (2019–2022), the level remained unvaried or decreased. This pattern agrees with the trend in MHWs in the western Mediterranean Sea. In fact, the significant sharp increase between 2015 and 2019 could be related to the intense MHWs that took place between 2015 and 2018. Iborra et al. [46] investigated the gorgonian population of the Gulf of Calvi (NW Corsica Island) over 15 years (2004, 2014, and 2019) and the observed changes in the level of necrosis of the colonies were related to the trend in MHWs occurring in that period. According to these authors, the effect of MHWs on the gorgonians reached relatively deep coastal waters (down to 40 m), where temperature increases are generally not recorded [46]. The rise in shallow water temperature induces a strong water column stratification, consequently resulting in low availability of trophic resources and high respiratory demand [56,57]. This hypothesis agrees with the bathymetric distribution of the level of epibiosis observed during our study in the Ligurian Sea, where the maximum values were recorded below 50 m. Previous diseases involving cnidarians and sponges in the same area were already observed to 70 m [17,58,59,60]. At the same time, the bathymetric distribution of the maximum level of epibiosis and entanglement (50–70 m) can also be related to the bathymetric distribution of *P. clavata* forests in the Ligurian Sea, being mainly concentrated between 45 and 65 m [47].

A change in the levels of epibiosis emerges from the present study. In the eastern Ligurian Sea, in four years (2015–2019), the percentage of epibionted colonies increased more than three times, and six times in the central one. This evidence indicates the damages inflicted by mass diseases following MHWs. In fact, during the 2003 episode, almost 80% of the studied colonies in the Gulf of Genoa developed epibiosis in a few weeks [21]. A long-term monitoring conducted on the *P. clavata* populations in the period 2003–2017 in the Scandola MPA highlighted a substantial decline in density and a slight loss of mean colony biomass after 15 years, and all populations were farther from recovery in 2017–2018 than in 2008. On average, in 2003, values decreased by around 71% and 80%, despite no significant changes in the size structure being observed in any population immediately after the 2003 MHW [45]. In contrast, Cupido et al. [59] observed a strong recovery of the populations in the La Spezia Gulf after the late-summer 1999 and 2003 large mortality events due to thermal stress. Long-term observations (1998 to 2008) confirmed that a positive net recruitment and canopy reestablishment could start some years after a high-mortality event under naturally stressed environmental conditions.

Despite this evidence, our data strongly suggest that human activities are involved in the development of epibiosis in *P. clavata*. The correlation, at the regional scale, between the level of epibiosis with the number of fishing vessels and harbours evidences the role of fishing activities. Moreover, the overlapping of the bathymetric trend in epibiosis with that of the entanglement indicates a strong correlation with this kind of impact. The depth range of epibionted colonies overlaps that of fishing with settled gear targeting spiny lobsters, sparids (common dentex, gilthead seabreams), and groupers [61]. Finally, the influence of fishing on the level of epibiosis was supported by the temporal correlation of the two parameters in all of the studied areas. In some cases (e.g., A3), it appears evident that a reduction in the entangled colonies anticipates that of epibionted ones.

These data strongly enforce the idea that the observed variations in the Ligurian forests are due to a synergistic effect of natural and anthropogenic causes. Probably, the damages caused by fishing activities accumulate on colonies already deeply stressed by thermal diseases. A further indication is that, in the period 2018–2022, despite the occurrence of one of the strongest MHW ever observed [55], the epibiosis remained stable or decreased concomitantly with a period of a strong reduction in the percentage of entanglement supporting a decrease in the fishing effort.

Unfortunately, data about the annual trend in artisanal fishing, and particularly recreative fishing, are virtually impossible to obtain, and therefore only a hypothesis can be formulated. We suggest that the reduction in the fishing activity speculated for the period 2019–2022 could be due to the COVID-19 pandemic lockdown imposed from March to May 2020. Fisheries were limited by the coronavirus pandemic at a global scale. In that period, the lockdown, followed by a strong shrinking in seafood demand, determined a decrease in fishing activities. For example, a recent study demonstrated a reduction of about 50% of the fishing effort for the Adriatic Sea [62]. In the eastern Mediterranean Sea, from December 2019 to February 2020, the average monthly gross margin for fishermen was 2.5 times less than the usual average [63]. The fishing effort in an exploited area of the north-western Mediterranean coast (Spain) during the lockdown dropped by 34%, landings were down by 49%, and revenues declined by 39% in comparison with the same period in 2017–2019 [64]. This scenario also well represents the conditions of the Ligurian fishing economy in that period, an economy based on traditional activities carried out by small artisanal communities, mainly targeting high-value resources [65].

In addition, during the lockdown period, the media reported the presence of iconic large marine animals, such as marine mammals, elasmobranchs, and marine turtles, in unexpected areas, such as close inshore areas or harbours [64]. Despite the wide interest in the effects of the COVID-19 pandemic on marine habitats and fisheries, no data are available about benthic organisms: our observation could be the first suggestion of an improvement in the health status of *P. clavata* forests resulting from a short-term strong reduction in fishing activities. The vulnerability of habitat-forming species is influenced by their low resilience to mechanical impacts driven by modest to slow growth rates [66]. This work supports the potential effectiveness of Fisheries Restricted Areas (FRAs) on these complex habitats. Finally, the impact of fishing cannot be separated from the potential source of damage produced by commercial harbours, which could be sources of pollution and marine litter [32,67]. Although no authors have put in evidence, a clear relationship between water quality and the level epibiosis on gorgonians, our data showed that the phenomenon is lower in A3 where the water quality index is slightly higher probably for the lack of important commercial harbour. Nevertheless, this datum has to be considered with due caution. In fact, the level of pollution is generally low in the entire Liguria coast [68].

As already stressed by several authors, the problems of the Mediterranean Sea are more related to habitat loss, overfishing, invasive species, etc., which do not affect the water quality at a water body level per se, although they can act at a more local scale [69].

## 5. Conclusions

In conclusion, the importance of long-term monitoring programmes are required to understand the trajectories of the modification of marine communities prone to different stressors. Despite the high effort, in terms of resources, required to obtain data over a long span of time, this kind of monitoring allows for the acquisition of fundamental information for conservation, providing evidence of unsuspected drivers of benthic community health, together with possible mitigation strategies. Moreover, the difficulty of discriminating the causative event of the epibiont colonisation affecting gorgonians, as well as the underestimation of damages produced by lost fishing gears not yet in place, may be overcome. Finally, the temporal sequence of images of single colonies could provide useful information about the age and development of the epibiontic community and the dynamic of the recovery processes by gorgonians.

## Figures and Tables

**Figure 1 biology-13-00642-f001:**
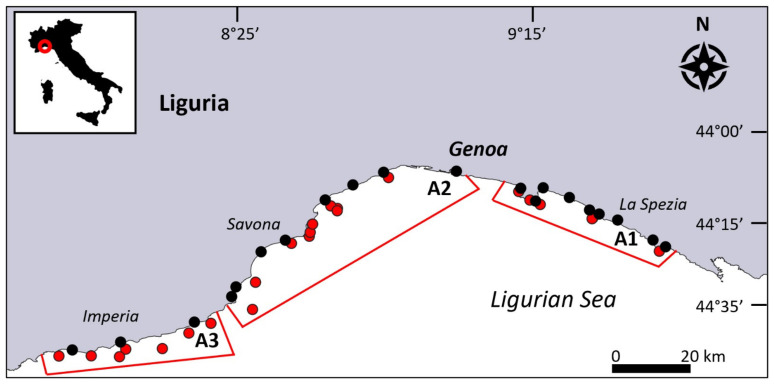
The Ligurian coast, with the localisation of the studied transects hosting the *Paramuricea clavata* forests (red dots), grouped into three areas. The black dots represent the fishing harbours.

**Figure 2 biology-13-00642-f002:**
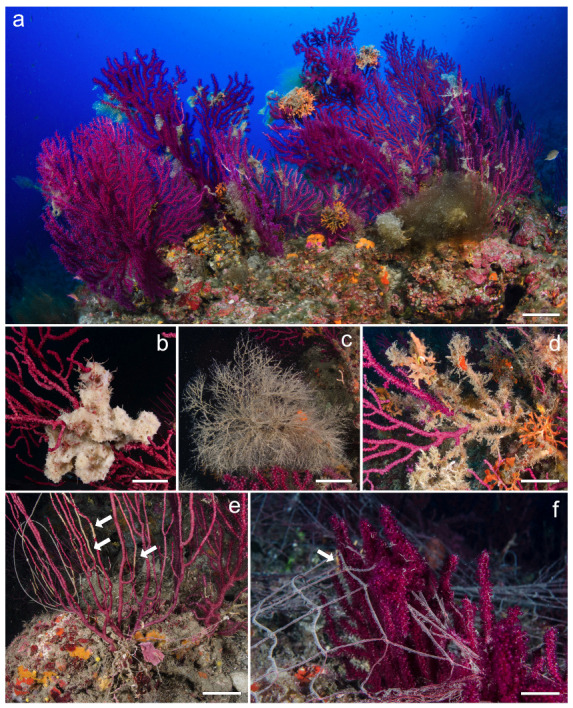
Examples of epibiosis and damages related to entanglements on *Paramuricea clavata*. (**a**) Several colonies show large masses of epibionts. Some typical epibionts such as (**b**) the encrusting sponge *Dysidea avara* (Schmidt, 1862), (**c**) hydrozoans, and (**d**) the branched bryozoan *Adeonella calveti Canu & Bassler*, 1930. (**e**) Colonies entangled in lost lines and (**f**) nets scraping portions of the coenenchime (white arrows). Scale bars: (**a**) 10 cm; (**b**–**d**) 2 cm; (**e**,**f**) 5 cm.

**Figure 3 biology-13-00642-f003:**
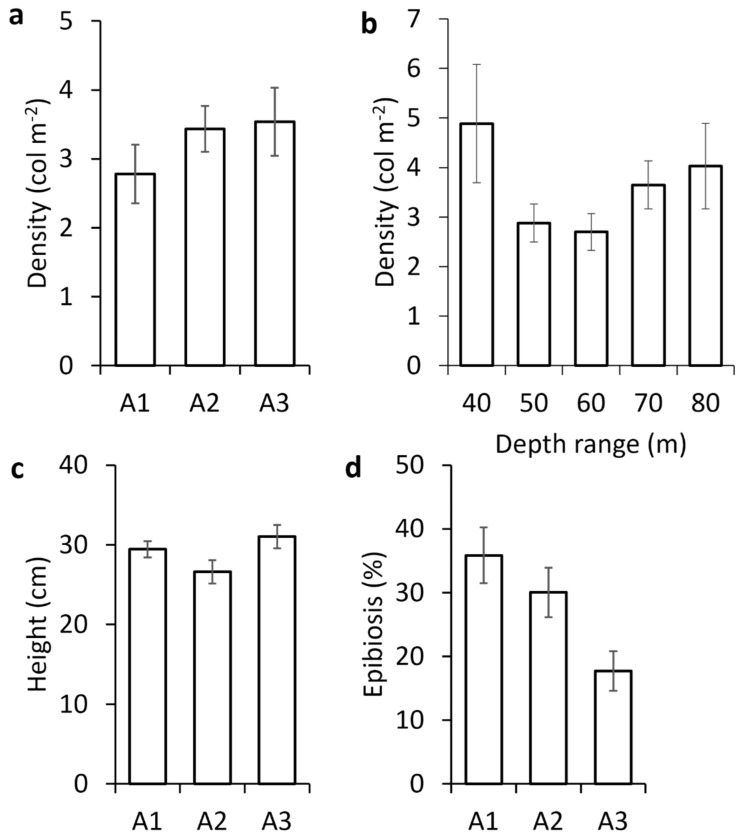
(**a**) Average density (±SE) in the three areas; (**b**) average density (±SE) in the different depth ranges; (**c**) average height (±SE), and (**d**) average percentage of epibionted colonies of *Paramuricea clavata* (±SE) in the three areas.

**Figure 4 biology-13-00642-f004:**
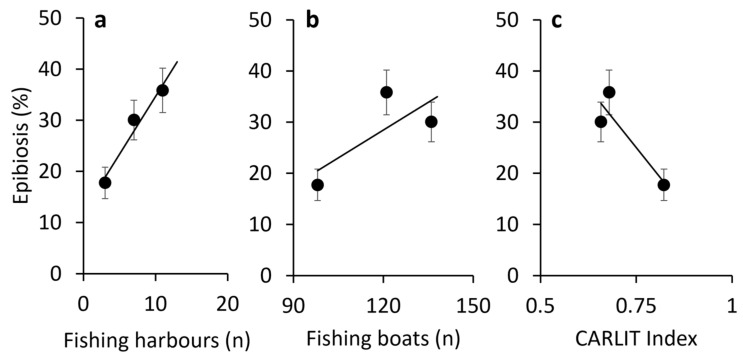
Correlation of the percentage of epibionted colonies in the three areas with the number of fishing harbours ((**a**), r = 0.98), fishing boats ((**b**), r = 0.75), and CARLIT Index ((**c**), r = 0.90) insisting on each area.

**Figure 5 biology-13-00642-f005:**
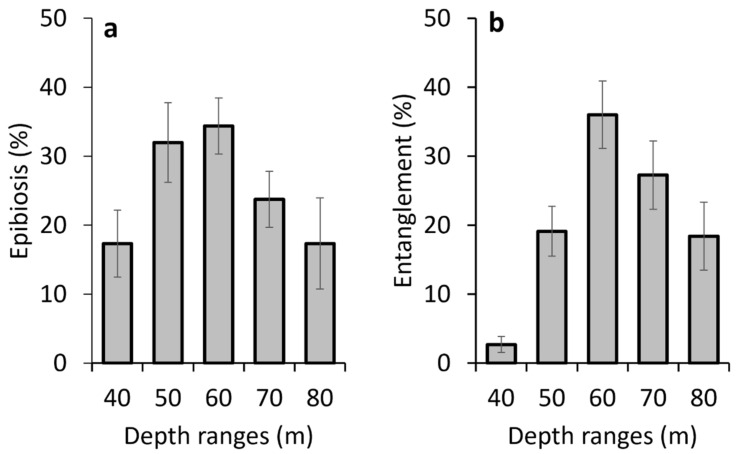
Bathymetric distribution of the percentage (**a**) of epibionted colonies and (**b**) of colonies entangled in lost fishing gear.

**Figure 6 biology-13-00642-f006:**
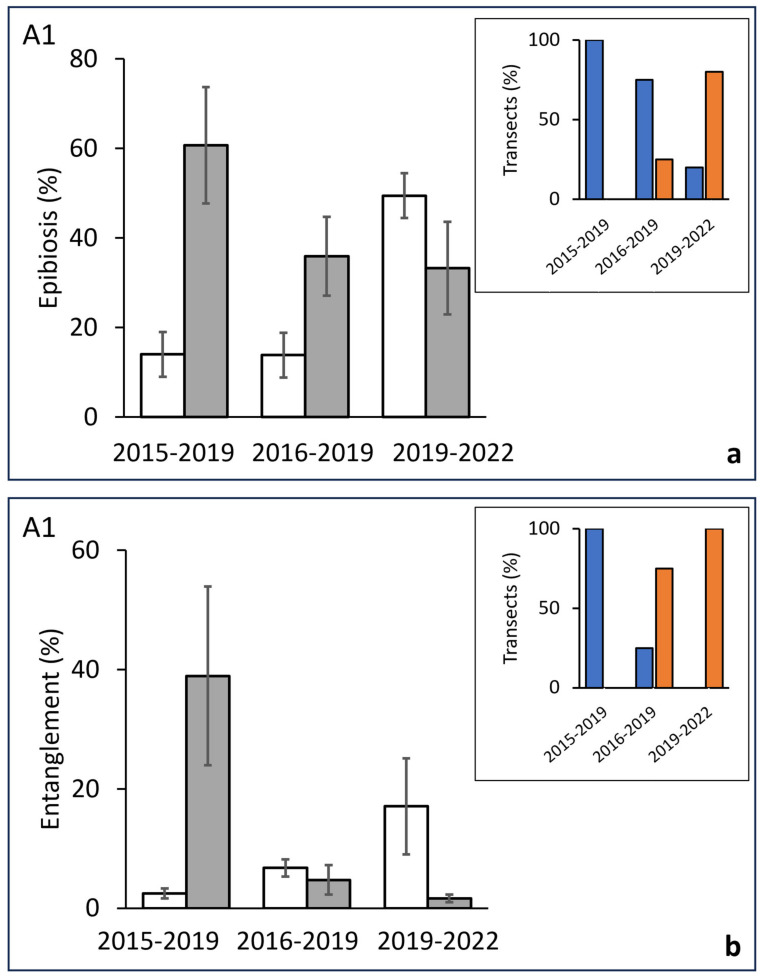
Eastern area (A1). (**a**) Differences across time of the percent of epibionted colonies (before, white bars/after, grey bars) in the same groups of transects replicated after 3–4 years. Inset: percentage of transects showing a positive (blue bars) or negative (orange bars) variation in the value of affected colonies in each considered period. (**b**) Differences across time (before, white bars/after, grey bars) of the percent of entangle colonies in the same groups of transects replicated after 3–4 years. Inset: percentage of transects showing a positive (blue bars) or negative (orange bars) variation in the value of entangled colonies in each considered period.

**Figure 7 biology-13-00642-f007:**
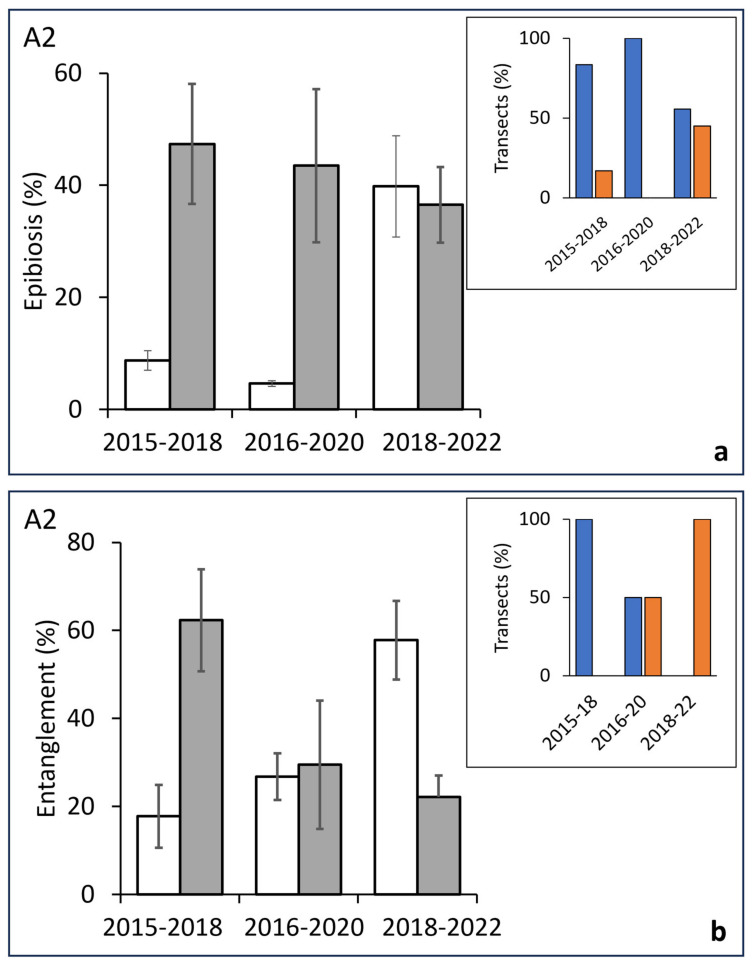
Central area (A2). (**a**) Differences across time of the percent of epibionted colonies (before, white bars/after, grey bars) in the same groups of transects replicated after 3–4 years. Inset: percentage of transects showing a positive (blue bars) or negative (orange bars) variation in the value of affected colonies in each considered period. (**b**) Differences across time (before, white bars/after, grey bars) of the percent of entangle colonies in the same groups of transects replicated after 3–4 years. Inset: percentage of transects showing a positive (blue bars) or negative (orange bars) variation in the value of entangled colonies in each considered period.

**Figure 8 biology-13-00642-f008:**
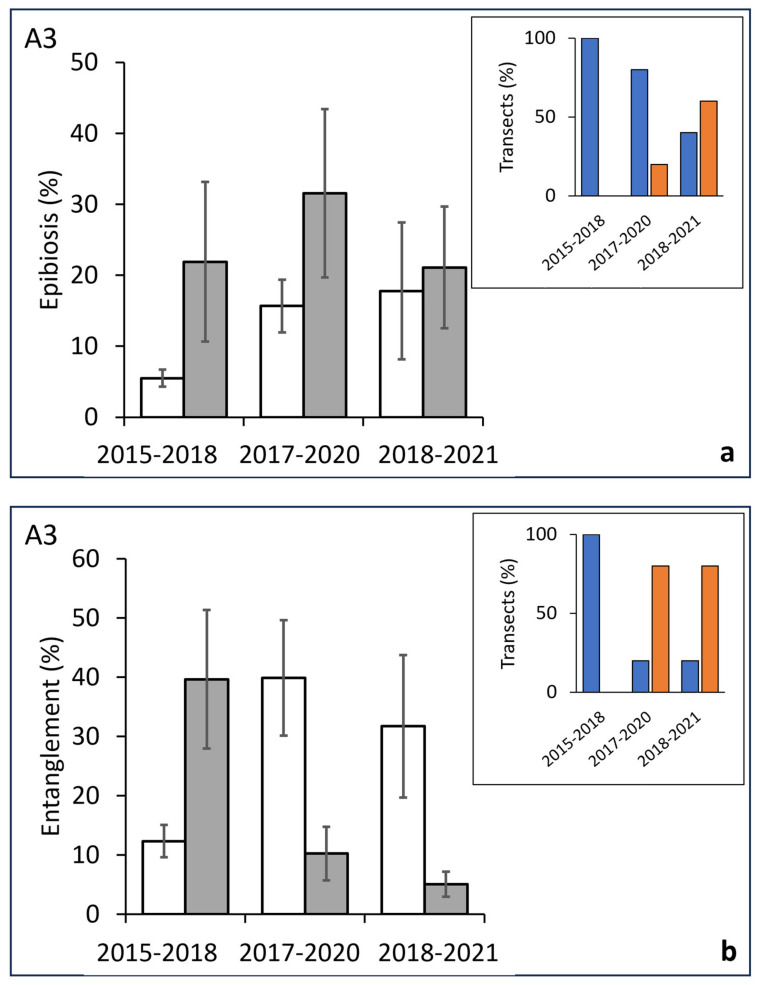
Western area (A3). (**a**) Differences across time of the percent of epibionted colonies (before, white bars/after, grey bars) in the same groups of transects replicated after 3–4 years. Inset: percentage of transects showing a positive (blue bars) or negative (orange bars) variation in the value of affected colonies in each considered period. (**b**) Differences across time (before, white bars/after, grey bars) of the percent of entangle colonies in the same groups of transects replicated after 3–4 years. Inset: percentage of transects showing a positive (blue bars) or negative (orange bars) variation in the value of entangled colonies in each considered period.

**Table 1 biology-13-00642-t001:** The repeated explored sites with the occurrence of *Paramuricea clavata* forests in the three areas of the Ligurian coast.

	Year	Site	Lat. (N)	Long. (E)	Transect ID	Depth (m)	N Colony	Density (col m^−2^)	Epibiosis %	Entanglement %	Av. H ± SE (cm)
Area A1	2016	Mesco Cape	44.13108	9.63407	PMMN_S2_T2	47	64	1.3	9.4	7.8	28.8 ± 2.3
	2019						66	0.5	15.9	4.5	20.3 ± 1.3
	2022						66	0.8	9.1	3.0	23.6 ± 0.9
	2015	Manara Cape W	44.243216	9.40207	SLMO_S3_T1	59	82	0.9	20.7	3.7	38.4 ± 2.2
	2019						107	4.1	40.0	77.6	35.5 ± 1.7
	2019				SLMO_S3_T3	41	198	2.0	73.0	21.2	30.2 ± 0.9
	2022						284	3.6	56.0	1.1	28.0 ± 1.1
	2015				SLMO_S3_T4	47	228	2.5	10.1	1.3	47.3 ± 2.2
	2019						95	1.4	89.4	46.3	35.0 ± 1.1
	2022						94	1.0	39.4	3.2	29.0 ± 1.7
	2015	Manara Cape E	44.24516	9.40409	SLMO_S2_T1	55	119	1.6	16.0	0.8	34.3 ± 2.3
	2019						212	2.6	76.0	22.2	35.3 ± 1.0
	2015				SLMO_S2_T2	64	97	1.0	9.3	4.1	32.2 ± 2.2
	2019						94	1.8	37.5	9.6	25.5 ± 1.1
	2016	Portofino Cape	44.2923	9.22313	AMPP_S1_T3	60	174	6.7	6.3	1.1	31.7 ± 1.2
	2019						115	1.2	58.8	9.6	32.7 ± 1.6
	2022						50	0.6	10.0	-	28.2 ± 1.3
	2016	Isuela Shoal	44.33713	9.14897	AMPP_S3_T2	33	838	10.6	24.6	2.7	24.2 ± 1.5
	2019						833	8.3	36.0	1.6	29.2 ± 1.0
	2022						649	6.6	8.8	-	30.8 ± 0.7
	2016				AMPP_S3_T3	52	332	3.8	15.1	6.9	24.9 ± 1.5
	2019						247	4.4	33.0	11.7	28.7 ± 1.5
	2022						104	1.2	53.1	1.0	28.5 ± 0.9
Area A2	2015	Arenzano-Varazze	44.38512	8.6996	NOAR_S1_T2	40	337	4.3	11.6	2.1	34.3 ± 1.3
	2018						411	5.1	60.3	42.8	33.2 ± 1.2
	2021						315	3.6	79.0	31.0	21.0 ± 0.9
	2015	Vado Ligure	44.2603	8.46638	NOAR_S2_T1	48	277	3.4	12.3	9.0	34.6 ± 2.0
	2018						207	4.0	28.5	50.7	31.1 ± 1.4
	2015				NOAR_S2_T2	63	184	1.5	12.5	14.2	30.0 ± 2.0
	2018						144	3.0	78.5	22.2	34.5 ± 1.2
	2021						152	3.2	44.0	8.0	22.3 ± 1.0
	2015				NOAR_S2_T3	56	72	0.9	5.6	4.2	13.4 ± 1.5
	2018						81	1.5	66.7	76.5	25.3 ± 1.2
	2016	Savona A	44.28739	8.50042	SVCL_S3_T2	45	291	3.5	3.8	22.0	29.8 ± 1.6
	2019						158	3.9	51.0	21.5	28.7 ± 0.9
	2022						314	4.5	7.6	9.2	25.0 ± 0.6
	2016	Savona B	44.27878	8.52335	SVCL_S2_T3	58	309	3.3	6.1	37.5	25.6 ± 1.7
	2019						125	2.6	76.7	72.0	35.3 ± 1.7
	2022						482	12.4	38.0	19.9	25.1 ± 0.8
	2015	Maledetti Shoal	44.22381	8.43657	NOAR_S4_T1	58	610	5.3	1.7	47.3	21.7 ± 1.0
	2018						276	2.8	42.4	84.4	25.2 ± 1.0
	2015				NOAR_S4_T2	68	69	4.2	8.8	29.8	17.8 ± 1.5
	2018						179	2.3	7.8	97.2	27.8 ± 1.2
	2021						89	2.1	28.2	12.8	12.5 ± 1.2
	2018				NOAR_S4_T4	56	271	3.2	42.1	87.8	29.8 ± 1.4
	2021						138	1.6	36.0	14.0	19.1 ± 0.7
	2018				NOAR_S4_T5	62	382	3.8	20.7	79.8	26.3 ± 1.3
	2021						352	4.2	30.0	58.0	21.5 ± 0.7
	2018				NOAR_S4_T7	53	68	0.7	10.3	36.8	15.8 ± 0.8
	2021						84	0.8	12.9	21.4	11.6 ± 0.5
	2018				NOAR_S4_T8	59	640	6.6	10.8	60.0	15.8 ± 0.4
	2021						421	4.2	53.0	25.0	12.5 ± 0.3
	2017	Finale Ligure	44.15817	8.36462	BONO_S1_T2	83	163	2.0	4.9	12.3	24.2 ± 1.2
	2020						112	5.9	34.3	18.8	33.2 ± 3.4
	2017				BONO_S1_T3	77	405	5.1	3.6	35.3	32.9 ± 1.2
	2020						453	5.3	12.0	5.7	65.7 ± 2.6
Area A3	2017	Albenga	44.02396	8.24063	ALGA_S3_T2	58	54	0.7	11.8	38.2	30.1 ± 1.5
	2020						72	1.2	70.8	1.4	35.4 ± 2.1
	2015	Diano Marina	43.88217	8.08675	SSDM_S1_T2	51	175	2.4	6.3	18.9	33.2 ± 1.5
	2018						119	2.0	55.5	68.9	39.4 ± 1.2
	2021						115	1.3	28.4	10.8	20.6 ± 1.0
	2018	Porto Maurizio	43.84858	8.01065	SSDM_S2_T3	36	76	1.2	1.3	-	37.8 ± 1.2
	2021						156	3.1	21.0	11.0	21.5 ± 0.9
	2017	Sanremo E	43.7929	7.79271	SRSST_S1_T2	69	265	3.3	18.1	46.2	27.7 ± 0.9
	2020						335	6.6	4.0	25.1	37.4 ± 1.3
	2017				SRSST_S1_T3	61	232	2.9	18.7	46.7	24.9 ± 1.2
	2020						135	3.5	7.0	26.7	31.7 ± 1.6
	2017	Sanremo W	43.76695	7.77113	BOSR_S3_T1	65	180	2.3	9.7	55.9	25.9 ± 1.1
	2020						181	4.4	10.4	1.7	38.3 ± 2.1
	2017				BOSR_S3_T2	49	54	0.7	9.7	55.9	32.9 ± 1.5
	2020						77	0.9	33.3	15.6	50.5 ± 2.2
	2016	Bordighera E	43.7699	7.67639	CMBO_S1_T1	49	155	1.9	4.5	5.8	19.5 ± 1.6
	2018						218	4.0	11.5	13.3	45.0 ± 1.2
	2021						556	8.4	7.0	3.4	21.7 ± 1.0
	2016				CMBO_S1_T2	60	179	1.9	8.4	13.4	26.8 ± 2.1
	2018						180	3.8	13.3	31.7	34.9 ± 1.6
	2021						823	13.1	49.0	6.0	24.8 ± 0.9
	2016				CMBO_S1_T3	66	400	5.0	2.8	11.3	22.5 ± 1.6
	2018						517	8.6	7.4	44.7	25.9 ± 1.2
	2021						407	5.1	0.0	0.0	13.6 ± 1.1
	2016	Mortola Cape	43.76952	7.56353	CMBO_S3_T2	35	302	3.8	29.1	3.3	28.1 ± 2.1
	2020						243	4.5	39.3	7.4	43.1 ± 1.9

**Table 2 biology-13-00642-t002:** Results of ANOVA and pair-wise comparisons of *Paramuricea clavata*, density, colony height, and percentages of epibionted specimens in each area. Bray-Curtis similarity index used for the resemblance matrix construction; permutation n = 9999. Significant effects are in bold.

	df	SS	MS	Pseudo-F	P (Perm)	Pair-Wises	T	P (Perm)
**Density**								
Area	2	5201.1	2600.5	2.716	0.3686			
Res	107	1.0245 × 10^5^	957.5					
Total								
**Height**						**Height**		
Area	2	1360.9	680.44	35.551	**0.0267**	A1 vs. A2	20.462	**0.0357**
Res	107	20,480	191.4			A1 vs. A3	0.7000	0.5049
Total	109	21,840				A2 vs. A3	22.188	**0.0249**
**Epibiosis**						**Epibiosis**		
Area	2	11,824	5912.2	38.355	**0.006**	A1 vs A2	12.312	0.1907
Res	107	1.65 × 10^9^	1541.4			A1 vs. A3	2.721	**0.0012**
Total	109	1.77 × 10^9^				A2 vs. A3	16.747	**0.0545**
**Entanglement**						**Entanglement**		
Area	2	25,589	12795	68.677	**0.0001**	A1 vs. A2	37.941	**0.0001**
Res	107	1.99 × 10^9^	1863			A1 vs. A3	19.837	**0.0119**
Total	109	2.25 × 10^9^				A2 vs. A3	16.131	**0.0553**

**Table 3 biology-13-00642-t003:** Results of PERMANOVA and pair-wise comparisons for the temporal analysis (before/after) of the percentage of epibiosis and entanglement of *Paramuricea clavata* in each replicated transect. Bray–Curtis similarity index used for the resemblance matrix construction; permutation n = 9999. Significant effects are in bold.

	df	SS	MS	Pseudo-F	P (Perm)	Pair-Wises	t	P (Perm)	Unique Perms	P (MC)
**A1**										
**Epibiosis**										
Before/After	5	11,612	2322.3	29.303	**0.0228**	2015/19	40.632	0.027	35	**0.0024**
Res	20	15,850	792.52			2016/19	20.984	0.0532	35	**0.0529**
Total	25	27,462				2019/22	0.79759	0.502	126	0.4923
**Entanglement**										
Before/After	5	18,518	3703.6	21.411	**0.0244**	2015/19	27.952	0.0259	35	**0.006**
Res	20	34,595	1729.7			2016/19	0.4955	0.7255	35	0.7313
Total	25	53,113				2019/22	14.721	0.0794	126	0.1193
**A2**										
**Epibiosis**										
Before/After	5	23,565	4713	46.103	**0.001**	2015/18	27.982	0.0135	462	**0.0043**
Res	32	32,713	1022.3			2016/20	38.243	0.0287	35	**0.0029**
Total	37	56,278				2018/22	0.516	0.709	8170	0.7162
**Entanglement**										
Before/After	5	15,630	3125.9	32.804	**0.0031**	2015/18	2.362	0.0203	461	**0.0157**
Res	32	30,493	952.92			2016/20	0.59803	0.8289	35	0.6893
Total	37	46,123				2018/22	30.371	0.0046	8150	**0.0043**
**A3**										
**Epibiosis**										
Before/After	5	9829.4	1965.9	12.736	0.2349	2015/18	18.318	0.0536	35	0.0779
Res	22	33,959	1543.6			2016/20	11.444	0.2483	91	0.2815
Total	27	43,788				2018/21	0.83113	0.6803	126	0.5607
**Entanglement**										
Before/After	5	17,345	3469	19.534	**0.0293**	2015/18	21.685	0.0855	35	**0.0435**
Res	22	39,068	1775.8			2016/20	15.939	0.1094	91	0.1148
Total	27	56,413				2018/21	15.329	0.0403	91	0.0872

## Data Availability

The authors confirm that the data supporting the findings of this study are available within the article. The video dataset collected and analysed during the present study is available from the corresponding author upon request.

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
