# Peer review of "Variation in the Health Status of the Mediterranean Gorgonian Forests: The Synergistic Effect of Marine Heat Waves and Fishing Activity"

_biology, 2024, doi:10.3390/biology13080642_

Round 1

Reviewer 1 Report

Comments and Suggestions for Authors

The manuscript entitled “Variation in the health status of the Mediterranean gorgonian forests: The synergistic effect of Marine Heat Waves and fishing activity” concerns a relatively little investigated topic and does this in an original and interesting way. The text is well written, Materials and Methods are detailed, the data in Results are generally presented clearly and the Discussion is consistent with the findings. It is ready to be published in the journal Biology after few small suggestions, which I list below.

In Mat and Met it is not clear whether the epibiosis is calculated as % of colony damage or as % of damaged colonies out of the total population.

In Results I suggest showing the colony densities also related to the bathymetric ranges, so that they can be better compared with the epibiosis and entanglement plots.

There are only few small typos in the text: in the materials there are some references written both as text and as numbers, some commas (like after Particularly at line 187) or conjunctions (like after 7 at line 204) missing, the font size for the caption of figure 7 is too large.

I suggest replacing or adding some more recent citations at line 69/70, such as:

Appolloni et al., 2020. β-Diversity of morphological groups as indicator of coralligenous community quality status. Ecological Indicators109, 105840.

Ferrigno et al.,2021. Diversity loss in coralligenous structuring species impacted by fishing gear and marine litter. Diversity13(7), 331.

Piazzi et al.,2022. Inconsistency in community structure and ecological quality between platform and cliff coralligenous assemblages. Ecological Indicators136, 108657.

Di Camillo et al., 2023 _Review of the indexes to assess the ecological quality of coralligenous reefs, towards a unified approach.

Author Response

REVIEWER 1:

We greatly appreciate the positive comments of REV. 1.

We thank you for the corrections and suggestions proposed, which we have tried to resolve as required.

Comments and Suggestions for Authors

The manuscript entitled Variation in the health status of the Mediterranean gorgonian forests: The synergistic effect of Marine Heat Waves and fishing activity” concerns a relatively little investigated topic and does this in an original and interesting way. The text is well written, Materials and Methods are detailed, the data in Results are generally presented clearly and the Discussion is consistent with the findings. It is ready to be published in the journal Biology after few small suggestions, which I list below.

In Mat and Met it is not clear whether the epibiosis is calculated as % of colony damage or as % of damaged colonies out of the total population.

ANSWER: we clarified it in the text

In Results I suggest showing the colony densities also related to the bathymetric ranges, so that they can be better compared with the epibiosis and entanglement plots.

ANSWER: OK

There are only few small typos in the text: in the materials there are some references written both as text and as numbers, some commas (like after Particularly at line 187) or conjunctions (like after 7 at line 204) missing, the font size for the caption of figure 7 is too large.

ANSWER: sorry for the mistakes, we checked and deleted the errors.

I suggest replacing or adding some more recent citations at line 69/70, such as:

Appolloni et al., 2020. β-Diversity of morphological groups as indicator of coralligenous community quality status. Ecological Indicators, 109, 105840.

Ferrigno et al.,2021. Diversity loss in coralligenous structuring species impacted by fishing gear and marine litter. Diversity, 13(7), 331.

Piazzi et al.,2022. Inconsistency in community structure and ecological quality between platform and cliff coralligenous assemblages. Ecological Indicators, 136, 108657.

Di Camillo et al., 2023 _Review of the indexes to assess the ecological quality of coralligenous reefs, towards a unified approach.

ANSWER: OK we added the references proposed.

Reviewer 2 Report

Comments and Suggestions for Authors

Review of Canessa et al, Variation in the health status.

This MS describes the results of ROV surveys of gorgonian forests off Italy. The data look solid, but the presentation is flawed. It appears that au’s minds were made up before the data were in, and it shows.

Using epibionts as a stress indicator is fine-after all, people have been doing this for decades. Mixing fishing damage in with epibionts seems…illogical. One is, we hope, a measure of water quality impacts on the gorgonians, whereas the other is a measure of extent of careless fishing. The connection with marine heat waves isn’t there-it exists only in au’s hopeful hearts.

The 500-kilo gorilla in the room makes its presence known in Fig 4. This plots the # of fishing harbours, and # fishing boats, against epibionts. Look at the correlations. # epibionts vs fishing HARBOURS is tight-correlation with # FISHING BOATS far less so. Harbours are potent sources of all sorts of pollution, whereas fishing boats by themselves not so much. So a MS that purports to see a connection between global warming and coral health actually shows that local pollution is more responsible.

The sad part of this is, au had in their hands a golden opportunity. Most ROV’s have at least some sampling capability. Samples of the affected gorgonians would have provided water temperature records and, via 15-N analyses, assessments of land-based organics.

This MS, as written, needs to be rejected. Au then have two choices: resubmit that data, which are quite good, and purge the MS of all unsupported discussion of causes. Or, better: get some samples, and give us some answers, in a followup MS.

Some minor issues below:

L81-ARE lacking

L96 coralligenous???

L156 fewer than

Comments on the Quality of English Language

Review of Canessa et al, Variation in the health status.

This MS describes the results of ROV surveys of gorgonian forests off Italy. The data look solid, but the presentation is flawed. It appears that au’s minds were made up before the data were in, and it shows.

Using epibionts as a stress indicator is fine-after all, people have been doing this for decades. Mixing fishing damage in with epibionts seems…illogical. One is, we hope, a measure of water quality impacts on the gorgonians, whereas the other is a measure of extent of careless fishing. The connection with marine heat waves isn’t there-it exists only in au’s hopeful hearts.

The 500-kilo gorilla in the room makes its presence known in Fig 4. This plots the # of fishing harbours, and # fishing boats, against epibionts. Look at the correlations. # epibionts vs fishing HARBOURS is tight-correlation with # FISHING BOATS far less so. Harbours are potent sources of all sorts of pollution, whereas fishing boats by themselves not so much. So a MS that purports to see a connection between global warming and coral health actually shows that local pollution is more responsible.

The sad part of this is, au had in their hands a golden opportunity. Most ROV’s have at least some sampling capability. Samples of the affected gorgonians would have provided water temperature records and, via 15-N analyses, assessments of land-based organics.

This MS, as written, needs to be rejected. Au then have two choices: resubmit that data, which are quite good, and purge the MS of all unsupported discussion of causes. Or, better: get some samples, and give us some answers, in a followup MS.

Some minor issues below:

L81-ARE lacking

L96 coralligenous???

L156 fewer than

Author Response

The comments raised by the Reviewer appear, in our humble opinion, to be inadequate to the judgment of the manuscript. We appreciate his work, but we have to disagree with these opinions and we provide an appropriate response.

This MS describes the results of ROV surveys of gorgonian forests off Italy. The data look solid, but the presentation is flawed. It appears that au’s minds were made up before the data were in, and it shows.

Using epibionts as a stress indicator is fine-after all, people have been doing this for decades. Mixing fishing damage in with epibionts seems…illogical. One is, we hope, a measure of water quality impacts on the gorgonians, whereas the other is a measure of extent of careless fishing. The connection with marine heat waves isn’t there-it exists only in au’s hopeful hearts.

The 500-kilo gorilla in the room makes its presence known in Fig 4. This plots the # of fishing harbours, and # fishing boats, against epibionts. Look at the correlations. # epibionts vs fishing HARBOURS is tight-correlation with # FISHING BOATS far less so. Harbours are potent sources of all sorts of pollution, whereas fishing boats by themselves not so much. So a MS that purports to see a connection between global warming and coral health actually shows that local pollution is more responsible.

The sad part of this is, au had in their hands a golden opportunity. Most ROV’s have at least some sampling capability. Samples of the affected gorgonians would have provided water temperature records and, via 15-N analyses, assessments of land-based organics.

This MS, as written, needs to be rejected. Au then have two choices: resubmit that data, which are quite good, and purge the MS of all unsupported discussion of causes. Or, better: get some samples, and give us some answers, in a followup MS.

ANSWER: Unfortunately, we cannot share the general comment of Reviewer 2. The reviewer in his criticism believes that we began this work having in mind that epibiosis is determined by mechanical damage from fishing and heat waves. The referee is absolutely right. We declare this idea already in the introduction. Dozens of papers published in the last 30 years demonstrate this statement. It has been shown photographically how the friction of the fishing lines on the gorgonian fans produces an abrasion of the coenenchyme on which the epibionts settle. Similarly, a wide literature demonstrates how heat waves affect gorgonians and favor epibiosis.
On the other hand, to our knowledge, there are no papers that relate epibiosis to levels of marine pollution which is very modest in the Ligurian Sea.
The image of the 500kg gorilla in the room that makes its presence known seems to suggest our intellectual dishonesty. This is false and, we might add, a little offensive.

On the other hand, the study aims not to explain the causes of epibiosis (they are already well known) but to demonstrate its quantitative variations in the short term. On this basis, we cannot agree with the referee by making the requested changes.
The minor corrections suggested were all accepted

Some minor issues below:

L81-ARE lacking

ANSWER: corrected.

L96 coralligenous???

ANSWER: coralligenous reefs.

L156 fewer than

ANSWER: mismatched

Reviewer 3 Report

Comments and Suggestions for Authors

Good job, well managed.

Methods adequate, only critics lie in the uncomplete covering of references. Many similar works, some with the same conclusions, other not, are not cited, for example that of the team of Boudouresque and co, other performed in the Scandola reserve or in the Calvi area in Corsica.

Clear and convincing.

Author Response

We would like to thank Reviewer 3 for the positive comment on our manuscript. We have added references in the text, as requested by other reviewers and we hope it will be appreciated.